# Analysis of the Athletic Career and Retirement Depending on the Type of Sport: A Comparison between Individual and Team Sports

**DOI:** 10.3390/ijerph17249265

**Published:** 2020-12-11

**Authors:** Cristina López de Subijana, Larisa Galatti, Rubén Moreno, Jose L. Chamorro

**Affiliations:** 1Departamento de Ciencias Sociales de la Actividad Física, Universidad Politécnica de Madrid, 28031 Madrid, Spain; c.lopezdesubijana@upm.es; 2Faculdade de Ciências Aplicadas, Limeira, Universidad de Campinas, São Paulo 13083970, Brazil; lgalatti@unicamp.br; 3Faculty of Sport Sciences, Universidad Europea de Madrid, C/Tajo s/n, Villaviciosa de Odón, 28670 Madrid, Spain; ruben.moreno2@universidadeuropea.es

**Keywords:** athletic career, sports career, elite sport, performance, dual career, employment, lifestyle

## Abstract

The type of sport practiced may shape the athletic career, considered as the period in which an athlete is dedicated to obtaining their maximum performance in one or more sports. The aim of this study was to compare athletic careers and retirement in individual and team sports. Four hundred and ten former elite athletes (38.5 ± 7.6 years) answered an ad hoc questionnaire; 61.5% were men and 38.5% women; 45.1% were from individual sports, while 54.9% were from team sports. It emerged that the age of maximum sports performance and the retirement age occurred significantly later in team sports than in individual sports (*U* = 15,042 and *U* = 12,624.5, respectively *p* < 0.001). Team sports athletes combined their athletic career with work to a greater extent than those from individual sports (*χ2* (3, *N* = 408) = 14.2; *p* = 0.003; *Cv* = 0.187). Individual sports athletes trained more hours per week (30.0 ± 11.7 h) than those involved in team sports (19.2 ± 10.7 h; *U* = 9682; *p* < 0.001). These athletes (team sports) were in a better economic and working situation at retirement transition (*χ2* (3, *N* = 406) = 23.9; *p* < 0.001; *Cv* = 0.242). Individual sports athletes perform physical activity more frequently than team sports athletes (*U* = 16,267.5; *p* = 0.045), while team sports athletes participate more actively in veteran competitions (*χ2* (1, *N* = 390) = 3.9; *p* = 0.047; *Cv* = 0.104) and more frequently attend events as spectators (*χ2* (1, *N* = 390) = 8.4; *p* = 0.004; *Cv* = 0.151). dual career support providers should be aware that team sports athletes enjoy a longer athletic career, and they are in a better position to face the retirement transition than individual sports athletes.

## 1. Introduction

The athletic career, considered as the period in which an athlete is dedicated to obtaining their maximum performance in one or more sports [1], is characterized by different stages: initiation, development, mastery (elite) and retirement [2,3]; although each sport has its own idiosyncrasies and the length of the stages and the ages differ very much from one sport to another [4]. Thus, the type of sport may shape the features of the athlete’s career.

The holistic model of athletic career development considers the athlete as a unique entity [5]. Hence, while the athlete develops the athletic career, he/she is developing as a person, developing in his/her relationships with others, and in some cases also developing a second career outside of sport. This model considers that each athlete has different spheres of life (sport, psychological, psychosocial, vocational, and financial) and postulates that these dimensions interact with one another. Thus, the dimensions should not be analyzed separately, and the athlete should be studied as a whole. In addition, throughout their athletic career, an athlete experiences transitions in each life sphere. Transitions are known as “a change in assumptions about oneself and the world and thus requires a corresponding change in one’s behavior and relationships” [6] (p. 5). These transitions can be normative (expected) or non-normative (unexpected) [1]. Normative transitions are the changes the athlete may expect to experience at the different stages of an athletic career; for example, entering a high-performance center after being selected for the national team. On the other hand, non-normative transitions are those that are unexpected. These could be injuries or a sudden change of coach. Facing non-normative transitions requires an effort to cope and deal with unwanted change. Stambulova’s model of sports transitions highlights that the balance between barriers and athletes’ resources mark successful or unsuccessful transition coping. When a transition is not overcome, a crisis could be experienced by the athlete followed by negative consequences such as being dropped, substance abuse or frustration [7].

Life spheres can interact with each other positively or negatively. In this respect, in the last decade, the studies on combining sport with a second career (studies or work) have increased in quantity as well as quality, exploring the benefits for the athletic career [8]. This combination is known in the European Union as developing a dual career (DC). In a recent study, Stambulova and Wylleman [8] reviewed state of the art on this topic. Developing a DC implies a strong commitment to both careers. Athletes that follow a DC deal with tight schedules, and they often mention their personal sacrifices, such as having little time for themselves, for personal relationships, for developing a social life, and for resting, in comparison with their counterparts. This situation may lead to stress, fatigue, injuries, and burnout [9,10,11]. On the other hand, once the athlete has adapted to the DC, the advantage is the perception of experiencing an optimal DC, balancing the different spheres of life: sport, a second career and personal life, where the athlete has the sense of having a “safety net” [8,12,13,14].

There are different ways of combining the athletic career with a second career. In a recent update, Torregrossa et al. [15] describe four different career paths. The *linear* path is when the athlete focuses exclusively on sport; the *convergent* path is when the athlete maintains the second activity (studies or work), but he/she gives priority to sport; the *parallel* path is when the two activities are combined, but the priority is shifted depending on external demands, and the athletic career is not always first; the *divergent* path is when the two careers put pressure on the athlete in a way that forces him/her to quit one of them (sport or studies/work). Managing the demands of both careers is not an easy task. While each sphere in life (sport, psychological, psychosocial, vocational, and financial) has its own normative transitions such as entering secondary school, entering the high-performance center, deciding whether to study at university or not, or whether to start working after achieving a degree, athletes create their own paths based on the decisions they make [8,16]. In order to prepare for each normative transition, athletes gather information from their peers/teammates about competition, or a training camp, or even about their skills for managing their studies and planning for their next steps. Planning has also been highlighted as a key element for coping satisfactorily with the sports retirement transition [17,18]. In addition, the lack of planning could be compensated with other resources, external or internal. External resources are like having unconditional support from a close social network [17,19,20,21,22,23]. For example, Willard and Lavalle [23] describe in a study with professional dancers how keeping the support circle intact during their whole career was a fundamental pillar in each transition. In relation to the internal resources, the education or academic level achieved is one of them [24]. Previous studies on this topic reinforce the idea that former elite athletes have a higher academic level than the general population [25,26]. In addition, in another study on elite athletes, women from individual sports reached a higher educational level than their counterparts [27]. However, in the comparison of individual and team sports athletes, the former are more likely to drop-out from secondary school [28]. This could be related to the hours of training per week, as the training load is greater in individual sports compared to team sports [29].

An active lifestyle is also desirable for the satisfaction of the retired athlete [30]. To our knowledge, studies regarding the physical activity performed by former elite athletes are scarce. Bäckmand et al. [31] reported in a sample of Finnish former elite athletes that 46% of them were physically active while 13% were sedentary. Torregrossa et al. [30], analyzing another large sample of former elite athletes from Colombia and Spain, found 54.6% were physically active and 12.9% sedentary. Both studies are based on self-reports, which usually under or overestimate the amount of Physical Activity in comparison with objective measures [32]. The recommendations of global organizations such as the World Health Organization (WHO, [33]) are clear in terms of quantity and intensity. They recommend adults aged 18–64 should do at least 150 min of moderate-intensity aerobic physical activity throughout the week or at least 75 min of vigorous-intensity aerobic physical activity throughout the week, or an equivalent combination of moderate- and vigorous-intensity activity. However, there is also a perception among this population of former athletes that accomplishing the recommended daily activities by going for a walk or cycling to work is not physical activity [34].

This research is in line with the recent review on DC by Stambulova and Wylleman [8], in which they state that one of the major gaps on this topic is the absence of training for the DC support providers. The purpose of this study is to compare the athletic career and retirement of former elite athletes according to the sport practiced and to provide information for the athletes’ counselors. Furthermore, describing the retirement transition of former elite athletes may help practitioners to aid the next generation of athletes at some key points to deal successfully with their transitions. Therefore, it could be interesting to analyze in-depth the athletic careers of former elite athletes and shed some light on their retirement and current lifestyles.

## 2. Material and Methods

### 2.1. Participants

The sample in this study was of 410 athletes (38.5 ± 7.6 years), 252 men (61.5%) and 158 women (38.5%). Their minimum time since retirement was 1 year (8 ± 6 years). Participants were former elite athletes [35] from 32 different Olympic sports [36]. They were all Caucasians. Forty-five percent (*n* = 185) were from individual sports while 54.9% (*n* = 225) were from team sports.

### 2.2. Measures

The submitted questionnaire had 55 items and followed the maximum information, minimum discomfort (MIMO) methodology [37]. In the first stage, the research group selected the questions related to the topic under study from two previous questionnaires: The social and working integration questionnaire and the Spanish version of the athlete retirement questionnaire [38,39,40]. In the second stage, a panel of experts with more than 15 years’ research experience in sports sciences, research methodology, and elite sport analyzed the pertinence of the questions [41,42]. This second version of the questionnaire had 55 items. A pilot study with 15 former elite athletes evaluated the understanding and clarity of each question and the adequate length of the questionnaire [43]. The final version of the questionnaire had five dimensions: the sociodemographic profile, sports profile, academic profile, employment, the retirement process, and their current lifestyle [44].

In this article, 23 variables were considered: gender, the type of sport, the events of the career path (age of starting practicing the sport, age of entering the elite level, and age of maximum sport performance), the hours the athlete trained per week at the elite level), their career path during their mastery stage (1 = solely devoted to sport-*linear* path; 2 = combined sport and education giving priority to sport-*convergent* path; 3 = combined sport and education and depending on the external demands-*parallel* path; 4 = combined sport and work); the sport retirement features, if the sport retirement process was planned, gradual and voluntary (1 = Yes; 2 = No, for each feature); if their working and economic situation was solved at retirement (1 = Yes, it was completely solved; 2 = Yes, it was mostly solved; 3 = No, I had only occasional jobs and 4 = No, I had hardly anything); the athletes’ level of studies at their retirement (1 = No higher education; 2 = Higher education); their working status (1 = Working; 2 = Not working, but I am looking for a job; 3 = Not working, and I am not looking for a job; and their monthly salary (1 = less than 1499€; 2 = 1500–2499€; 3 = over 2500€); their relation with sport nowadays (I do physical activity or practice sport; I compete in veterans events; I keep in touch with my coaches; I have an employment related to sport; I attend sport events as a spectator; I informally counsel young athletes) as (1 = yes; 2 = no); if they practiced sport or physical activity, they answered about the frequency (1 = once a week or less; 2 = 2–3 times per week; 3 = more than 3 times per week) and how many hours they spent each session (1 = 1 h or less; 2 = between 1 and 2 h; 3 = more than 2 h).

### 2.3. Procedure

The former elite athletes were recruited through different stakeholders (Spanish Sports Council, national sports federations, and elite athlete associations) using snowball sampling [45]. Prior to participation in the study, the athletes confirmed that they were involved in it voluntarily and signed a consent form. The university ethics committee approved the study protocol (E15 11580 172).

### 2.4. Data Analysis

Data analysis was performed using SPPS version 26. In order to compare the differences between groups, nonparametric Pearson’s chi-squared and Mann–Whitney *U* tests were applied. The effect size indicator was Cramer’s V (*Cv*) and Eta Square (*η^2^*; [46]. In Pearson’s chi-squared tests with more than two levels of the variable, the standardized adjusted residuals were analyzed to report the significant differences at each variable level [47,48,49]. The significance level was set at α = 0.05.

## 3. Results

### 3.1. Athletic Career and Career Path

The most important events in the athletic career and its length are shown in Table 1. Both groups of athletes began their athletic careers at a similar age (10.7 ± 4.8 years for individual sports; 10.9 ± 3.8 years for team sports; *p* > 0.05). The individual athletes entered the elite level (17.5 ± 4.0) earlier than the team sports athletes (18.1 ± 2.5 years; *U* = 17,649.5; *p* = 0.005). This difference increased in the next parameter as the team sports athletes’ moment of maximum achievement (24.9 ± 4.5 years) occurred later (*U* = 15,042; *p* < 0.001) than those of the individual sports athletes (23.0 ± 5.4 years). Team sports athletes (31.8 ± 5.1 years) retired later than individual sports athletes (27.7 ± 6.8 years; *U* = 12,624.5; *p* < 0.001), too. Consequently, the length of the elite stage was longer in team sports athletes (13.8 ± 4.9 years) than individual sports athletes (10.3 ± 5.3 years; *U* = 12,493; *p* < 0.001). In addition, the length of the athletic career in team sports was longer (20.9 ± 5.8 years; *U* = 13,496; *p* < 0.001) than the individual sports athletes (17.0 ± 6.4 years).

Regarding the career path during their mastery (elite) stage, 15% of the athletes were solely devoted to sport, 64.9% combined sport with studies and 20.1% combined sport with work (Figure 1). Significant differences were found in the career path in the group comparison (*χ2* (3, *N* = 408) = 14.2; *p* = 0.003; *Cv* = 0.187). Team sports athletes combined their athletic career with work to a greater extent (26.8%; adj. res. = 3.7) than those from individual sports (12.0%). Athletes involved in individual sports trained more hours per week (30.0 ± 11.7 h) than those involved in team sports (19.2 ± 10.7 h; *U* = 9682; *p* < 0.001).

### 3.2. Sport Retirement

By the time of retirement, there were significant differences between both groups in the level of studies reached (*χ2* (1; *N* = 409) = 4.9; *p* = 0.027; *Cv* = 0.109; see Table 2). The team sports athletes had more frequently reached higher education (58.2%) at the end of their athletic career than the athletes practicing individual sports (47.3%). Most of the former elite athletes in this study retired voluntarily (82.6%); radically (66.3%) and in an unplanned manner (59.5%). Team sports athletes planned their retirement more frequently (45.8%) than individual sports athletes (34.1% *χ2* (1; *N* = 410) = 5.8; *p* = 0.016; *Cv* = 0.119). Team sports athletes were in a better economic and working situation at this transition period than individual sports athletes (*χ2* (3; *N* = 406) = 23.9; *p* < 0.001; *η^2^* = 0.218) and were more frequently in a “mostly solved” situation (35.9%; adj. res. = 3.7) than individual sports athletes (19.1%). At the same time, individual sports athletes reported more frequently (48.6%; adj. res. = 4.3) that they had “hardly anything” at retirement than the team sports athletes (27.8%).

### 3.3. Employment and Current Lifestyle

The working status and relation with sport nowadays resultsa are shown in Table 3. Nowadays, regarding their employment status, 88.6% were employed, 9.7% were unemployed, while 1.6% did not need to work. In the type of sports comparison, the employment status and salary situation were similar (*p* < 0.05) for both groups of athletes. In their relationship with sport nowadays, significant differences appeared in two variables: competing in veterans’ events and attending sports events as spectators. Team sports athletes participated more actively in veteran competitions (39.4%) than individual sports athletes (29.4%; *χ2* (1; *N* = 390) = 3.9; *p* = 0.047; *Cv*= 0.104), and attended events as spectators more frequently (77.8%) than individual sports athletes (64.1%; *χ2*(1; *N* = 390) = 8.4; *p* = 0.004; *Cv* = 0.151). The sedentary rate of the former elite athletes in this study was 8.7%. This variable (I do physical activity or practice sport) did not show significant differences in the group comparison (*p* > 0.05). However, among the 91.3% that practiced physical activity and exercise in the group comparison, on a scale from 1 to 3, (1 = once per week or less; 2 = twice per week; 3 = 3 times per week or more), individual sports athletes reported that they practiced physical activity more frequently (2.2 ± 0.8) than team sports athletes (2.0 ± 0.8; *U* = 16,153.5; *p* = 0.033).

## 4. Discussion

The aim of this study was to compare the athletic career and retirement of individual and team sports athletes. The athletic career of the team sports athletes was longer than that of the individual sports athletes. Significant differences were found in the career path in the group comparison. Team sports athletes combined sport with work during the elite stage to a greater degree than the individual sports athletes. Individual sports athletes trained more hours per week than team sports athletes. By the time of their retirement transition, team sports athletes had finished higher education studies to a greater extent and were in a better economic and working situation than individual sports athletes, and they also planned more frequently the retirement transition. This sample’s employment status is better than that of the general Spanish population. There was no difference in salaries between both groups. Regarding their current lifestyles, the individual sports athletes performed physical activity more frequently than team sports athletes, but the latter competed more frequently in veterans’ events and attended sports events more frequently as spectators. We describe the relevance of these findings, highlight the limitations of this research, and propose practical implications in the following sections.

### 4.1. Athletic Career and Career Path

Our results regarding the events of the athletic career agree with previous studies: the age of beginning to practice the sport at around 10 years old, entering the elite level at 18 years old, and retirement at 30 years old, are in line with studies on German, Slovene or UK samples [28,50,51]. However, the contribution of this study is on the differences found in the type of sport comparing the length of the athletic career. Team sports athletes may need more time to achieve their maximum sports performance, but once they reach that peak, they stay longer at the elite level than individual sports athletes.

The career path chosen during the mastery stage is slightly different from the one reported regarding Spanish Olympic athletes [25]. In that study, 24% of the Olympic athletes were solely devoted to sport, 66.2% combined sport and studies and 9.1% combined sport and work. In our sample, 15% were solely devoted to sport and 20.1% combined sport and work. These differences could be due to the fact that this sample was of the athletes at the elite level named by the government while the Barriopedro et al. [25] sample were athletes that participated in the Olympic games. The Spanish elite athlete population ranges from 3500 to 5000 [34], while the Spanish Olympic delegation for the summer games is of about 300 athletes [35]. Hence, in this study, the sample had a lower sports level than that of Barriopedro et al. [25]. Participating in the Olympic games ensures that the athlete attains a Government scholarship at the highest level. Therefore, this may explain that in this study, the athletes devoted solely to the sport were less numerous, and moreover that they combined sport and work more frequently, due to the need for an income. In the comparison of the career path regarding the type of sport, more team sports athletes combined sport and work. Athletes of team sports had longer athletic careers, so they entered another stage at the financial point where they could combine their sport with a job. In addition, the individual athletes trained more hours per week, so their options to combine sport with a job were limited. Tekacv et al. [51] pointed out how one of the main barriers to combining sport and work is the lack of flexible timetables in both careers.

### 4.2. Sport Retirement

Regarding the features of the discontinuation or retirement transition, the positive point is that most of the athletes retired voluntarily; however, they did so radically. Radical retirement is not recommended as gradually decreasing the amount of training and competitions previously to retirement prepare the body and mind more smoothly for the next stages in life [52]. With respect to the type of sports comparisons, the team sports athletes were better prepared for this transition: they had higher education studies, they were in a better economic and working position, and they planned their sport retirement more frequently than the individual sports athletes. Following previous findings [16,17,18], team sports athletes have the internal (education and planning) and external (financial) resources to cope with the retirement transition more successfully than individual sports athletes.

### 4.3. Employment and Current Lifestyle

Concerning the employment status, our results show how the former elite athletes in this study had a better unemployment rate than the general population [53]. Another interesting finding is in relation to their current lifestyle. In comparison with previous studies [29,30] and with the general population (46%, [54]), this sample had a lower percentage of sedentary individuals (8.7%). It is noteworthy that the team sports athletes still engage in competitions and sports events as spectators more frequently than individual sports athletes. Torregrossa et al. [30] found competition was a reason that was related to intrinsic sports motivation in former elite athletes’ physical activity. In this study, these athletes (team sports) maintained that bond with sport, as for them, competition is part of the pleasure and fun of practicing sport, as opposed to the general population for whom competition is neither a pleasure nor fun [30].

### 4.4. Limitations

Overall, this research compares in detail the athletic career and retirement of individual and team sports athletes. Further, it describes their current employment and lifestyle. It is, however, important to recognize certain research limitations. First, as it is a cross-sectional study, no causal relationship could be established. Second, the sports disciplines were classified into two groups, so each sports cultural framework was not considered [12]. Third, the physical activity was assessed by means of recall; these data should be cross-checked with an objective measuring device (pedometer, accelerometer, etc. [31]). Moreover, finally, the snowball sampling technique did not allow us to confirm the identity of the athlete.

### 4.5. Practical Implications

The practical implications of this research lead us to propose some recommendations. DC support providers are those that can help sports athletes to prepare for the retirement transitions under better conditions. For example, based on the high training load of individual sports athletes, they should counsel them to engage in flexible educational programs. Moreover, team sports athletes seem to enjoy participating in competitions, so different recreational competitions could be proposed for former elite athletes to maintain physical activity and create a support network among themselves.

## 5. Conclusions

In summary, the findings of this study highlight the differences in the athletic career and retirement of individual and team sports athletes. Team sports athletes reach their maximum peak performance later than individual athletes; they enjoy a longer athletic career and combine their sport with work more frequently than individual sports athletes. At the time of retirement, the team sports athletes are in a better financial position than the individual sports athletes. In addition, Spanish former elite athletes’ lives nowadays, in terms of employment, are better than those of the general population, and they are also active and healthy. Sports stakeholders (National Olympic Committee, national government, sports federations and sports clubs) should be aware that individual sports athletes are in a worse position for facing the sports retirement transition than team sports athletes.

## Figures and Tables

**Figure 1 ijerph-17-09265-f001:**
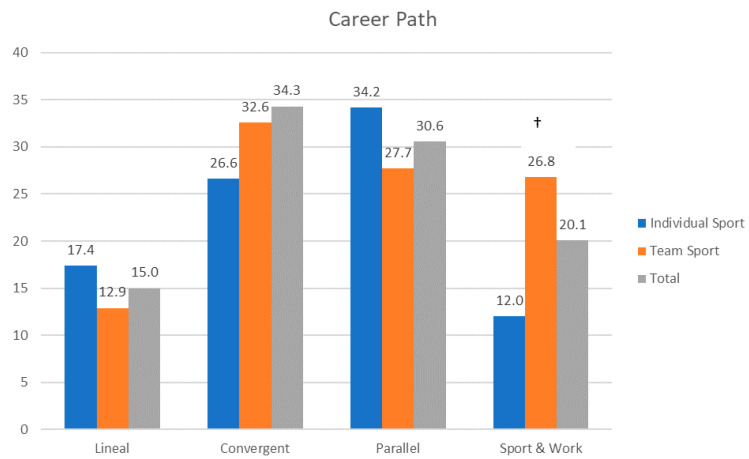
Athletes’ career paths regarding the type of sport in percent (note: ^†^ adjusted residuals above 3).

**Table 1 ijerph-17-09265-t001:** Events and length of athletic career regarding the type of sport in years.

	Individual Sport				Team Sport				Total			
	M	SD	95% CI		M	SD	95% CI		M	SD	95% CI	
Variable			LL	UL			LL	UL			LL	UL
Beginning to practice	10.7	4.8	10.0	11.4	10.9	3.8	10.5	11.4	10.8	4.2	10.4	11.3
Entering the elite level *	17.5	4.0	16.9	18.1	18.1	2.9	17.7	18.4	17.8	3.4	17.5	18.1
Maximum achievement **	23.0	5.4	22.2	23.7	24.9	4.5	24.3	25.5	24.0	5.0	23.6	24.5
Retirement **	27.7	6.8	26.8	28.7	31.8	5.1	31.2	32.5	30.0	6.3	29.4	30.6
Length elite stage **	10.3	5.3	9.5	11.1	13.8	4.9	13.2	14.4	12.2	5.3	11.7	12.8
Length athletic career **	17.0	6.4	16.1	18.0	20.9	5.8	20.1	21.7	19.2	6.4	18.5	19.8
Length in elite up to the maximum achievement **	5.4	3.9	4.9	6.0	6.9	4.2	6.3	7.5	6.2	4.2	5.8	6.6
Length from maximum achievement until retirement **	4.9	3.8	4.3	5.4	7.0	4.3	6.4	7.5	6.0	4.2	5.6	6.4

Note: M = mean; SD = standard deviation; CI = confidence interval; LL = lower limit; UL= upper limit; * *p* < 0.05, ** *p* < 0.001 for group comparisons.

**Table 2 ijerph-17-09265-t002:** Overview of retirement regarding the type of sport.

		Individual Sports	Team Sports	Total
Retirement Features		% (*n* = 185)	% (*n* = 225)	% (*n* = 410)
Timing retirement	Radical	64.9	67.6	66.3
	Gradual	35.1	32.4	33.7
Voluntariness	Voluntary	79.3	85.3	82.6
	Involuntary	20.7	14.7	17.4
Planned *	No	65.9	54.2	59.5
	Yes	34.1	45.8	40.5
Economic and working situation at retirement *	Completely solved	13.1	18.8	16.3
	Mostly solved	19.1	35.9 ^†^	28.3
	I had only occasional jobs	19.1	17.5	18.2
	I had hardly anything	48.6 ^†^	27.8	37.2

Note: % = column percentage; * *p* < 0.05 for group comparisons; ^†^ adjusted residuals above 3.

**Table 3 ijerph-17-09265-t003:** Working status and relation with sport nowadays regarding the type of sport.

		Individual Sport	Team Sport	Total
Employment Status and Salary		% (*n* = 185)	% (*n* = 225)	% (*n* = 410)
Working status	Yes, working	88.6	87.6	88.0
	Not working and looking for a job	9.7	9.3	9.5
	Not working and not looking for a job	1.6	3.1	2.4
		(*n* = 160)	(*n* = 180)	(*n* = 340)
Monthly income	Up to 1499 €	37.5	34.4	35.9
	1500–2499 €	41.9	37.8	39.7
	more than 2500 €	20.6	27.8	24.4
Relation with Sport Nowadays		(*n* = 176)	(*n* = 214)	(*n* = 390)
I do physical activity or practice sport	No	7.4	9.8	8.7
	Yes	92.6	90.2	91.3
I compete in veterans’ events *	No	70.6	60.6	65.3
	Yes	29.4	39.4	34.7
I keep in touch with my coaches	No	29.5	31.3	30.4
	Yes	70.5	68.7	69.6
I have a job related to sport	No	53.6	52.9	53.3
	Yes	46.4	47.1	46.7
I attend sports events as a spectator *	No	35.9	22.2	28.5
	Yes	64.1	77.8	71.5
I counsel young athletes	No	41.5	42.6	42.1
	Yes	58.5	57.4	57.9

Note: % = column percentage; * *p* < 0.05 for group comparisons.

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
