# Peer review of "Analysis of the Athletic Career and Retirement Depending on the Type of Sport: A Comparison between Individual and Team Sports"

_ijerph, 2020, doi:10.3390/ijerph17249265_

Round 1
Reviewer 1 Report
Dear Authors,
The reviewer would like to thank you for submitting your work for review. The purpose of this review is to improve the overall presentation of your work to aid in a successful publication. Please see comments below:
Line 54-56: “They also mention lack of time for themselves or for resting in comparison to their counterparts…., the advantages of being enrolled in a DC are the balanced life…” – This seems to contradict each other. Please elaborate to add clarity.
Line 66 – The transition to apply planning only towards the retirement of the athlete seems quite abrupt. Consider a smoother transition, maybe via incorporating the previous “divergent path” and how it relates to planning?
Line 83-84 – Please note that most reputable physical activity studies rely not only on recall and the limitation to subject interpretation as to what warrants physical activity characterization. Instead, physical activity research heavily relies on actigraphy, an objective measure of physical activity that only requires subjects to record times of non-wear, which can still be validated by looking at the data collected. For clarity, please discuss this limitation in the literature as it pertains to your sample population.
Line 86-88 – The purpose of the study is not clearly explained in this section. Please clarify the gap in literature and knowledge and how this study will add to the current knowledge. Clearly state your purpose of gathering the questionnaire data.
Figure 1. – Please adjust font sizes of in-figure-legends, so they will not overlap with the graphic.
Add a figure legend to explain what is depicted.
Table 3 – Please reformat the table, so it is easier to grasp information. The current depiction is hard on the eye.
Line 262-279 - You specifically conclude how the data presented, and your interpretation of the findings could improve athlete career transition and motivate program designs for sport stakeholders. Please use this in your introduction to convince the audience of the gap in the literature. This will add and value to the study and justify its publication.
The reviewer would like to thank the authors again for their work and is looking forward to the addressed responses.
Author Response
REVIEWER 1
Dear Authors,
The reviewer would like to thank you for submitting your work for review. The purpose of this review is to improve the overall presentation of your work to aid in a successful publication.
Dear editor and reviewers:
We really appreciate the effort you made in reviewing our manuscript. Thank you very much for the suggestions made on the article as they certainly helped to improve it. We hope now the article is structured and written more correctly. We have addressed each comment and suggestion and have detailed the specific changes in the following document. Within the manuscript itself, we have typed the changes in red. We have reviewed the quality of the English writing again with a professional translator. We hope this version is suitable for publication.
Thank you for everything, best regards,
The authors
Please see comments below:
Line 54-56: “They also mention lack of time for themselves or for resting in comparison to their counterparts…., the advantages of being enrolled in a DC are the balanced life…” – This seems to contradict each other. Please elaborate to add clarity.
Authors’ answer: Thank you for your comment. We have rewritten this part of the paragraph; lines 57-63. We hope it is clearer now.
Line 66 – The transition to apply planning only towards the retirement of the athlete seems quite abrupt. Consider a smoother transition, maybe via incorporating the previous “divergent path” and how it relates to planning?
Authors’ answer: Thank you for the suggestion. We have rewritten this part in order to link planning with the divergent path more smoothly; lines 70-76.
Line 83-84 – Please note that most reputable physical activity studies rely not only on recall and the limitation to subject interpretation as to what warrants physical activity characterization. Instead, physical activity research heavily relies on actigraphy, an objective measure of physical activity that only requires subjects to record times of non-wear, which can still be validated by looking at the data collected. For clarity, please discuss this limitation in the literature as it pertains to your sample population.
Authors’ answer: Thank you for this constructive criticism. We have added some information about the measures relying on recall rather than objective measures. We have also added the Physical activity recommended by the World Health Organization; lines 91-97.
Line 86-88 – The purpose of the study is not clearly explained in this section. Please clarify the gap in literature and knowledge and how this study will add to the current knowledge. Clearly state your purpose of gathering the questionnaire data.
Authors’ answer: Thank you for the suggestion. We have related the practical implications with the purpose of the study and with DC support providers’ need for information in relation to their work with athletes; in lines 102-107.
Figure 1. – Please adjust font sizes of in-figure-legends, so they will not overlap with the graphic.
Add a figure legend to explain what is depicted.
Authors’ answer: Thank you for the suggestion. We have added the legend on the right side of the image. We have also adjusted the font size of in-figure legends.
Table 3 – Please reformat the table, so it is easier to grasp information. The current depiction is hard on the eye.
Authors’ answer: Thank you for your comment. We have added some labels in bold and the number of athletes that answered each question. We hope the tables are easier to read now.
Line 262-279 - You specifically conclude how the data presented, and your interpretation of the findings could improve athlete career transition and motivate program designs for sport stakeholders. Please use this in your introduction to convince the audience of the gap in the literature. This will add and value to the study and justify its publication.
Authors’ answer: Thank you for your suggestion. We have added some information about the gap in the previous literature to help readers understand the meaning of this research, in lines 102-107.
The reviewer would like to thank the authors again for their work and is looking forward to the addressed responses.
Authors’ answer: Thank you again for your work and interest in reviewing the manuscript.

Reviewer 2 Report
In the article entitled “Analysis of the athletic career depending on the type of sport: a comparison between individual and team sports”, the authors aimed to compare the athletic careers of athletes involved in individual and team sports.
The manuscript proposes an interesting topic, but it is poor written and most of the parts need modifications.
Abstract
The authors do not follow a conceptual flow. They begin the paragraph stating the aim of the study, and neither a background nor an introduction was reported.
Please, reformulate the abstract section.
Introduction
The introduction is set up in an incorrect manner. For example, the authors insert the aim of the study at the beginning of the section. In addition to conceptual errors, also typing mistakes are present (e.g. line 32, line 81).
Please, the introduction should be reformulated in a better way, in order to provide both, a correct and essential background of the topic and to permit the readers to clearly understand the contents.
The section 2 should be the Materials and Methods section.
At line 103, the authors state that a pilot study was performed. Please, include the reference to provide the results of the pilot study.
Results
In the section 3.1, authors reported the Mann Whitney test in an incorrect manner. Please, report the results in the correct way, following an existing style.
At the same way, the chi-square test was incorrectly reported. Please, write the chi-square results in the correct manner.
The description of Figure 1 is missing. Please, the authors should explain better what the figure reports and the meaning of the asterisk.
In the following part of the results section, the authors report the chi-square results in an incoherent manner. Please, uniform the style used to report statistical results.
In section 4.3, the authors used interchangeably the concept of exercise and physical activity, while they are different. In addition, please less speculation should be done on results unsupported by data.
The remaining part of the manuscript is poor written and includes typing errors. Please, reformulate in a correct manner.
Finally, in the affiliation section, the instruction provided from the editor are not respected.
Author Response
REVIEWER 2
In the article entitled “Analysis of the athletic career depending on the type of sport: a comparison between individual and team sports”, the authors aimed to compare the athletic careers of athletes involved in individual and team sports.
The manuscript proposes an interesting topic, but it is poor written and most of the parts need modifications.
Authors’ answer:
Dear editor and reviewers:
We really appreciate the effort you made in reviewing our manuscript. Thank you very much for the suggestions made on the article as they certainly helped to improve it. We hope now the article is structured and written more correctly. We have addressed each comment and suggestion and have detailed the specific changes in the following document. Within the manuscript itself, we have typed the changes in red. We have reviewed the quality of the English writing again with a professional translator. We hope this version is suitable for publication.
Thank you for everything, best regards,
The authors
Abstract
The authors do not follow a conceptual flow. They begin the paragraph stating the aim of the study, and neither a background nor an introduction was reported.
Authors’ answer: Thank you for your comment. We have added two sentences as theoretical background in the abstract, in lines 14-16.
Please, reformulate the abstract section.
Introduction
The introduction is set up in an incorrect manner. For example, the authors insert the aim of the study at the beginning of the section. In addition to conceptual errors, also typing mistakes are present (e.g. line 32, line 81).
Authors’ answer: Thank you for your comment. We have deleted the sentence mentioning the aim of the study at the beginning of this section.
We have also inserted the first paragraph in the introduction, in lines 33-37.
We have reviewed the manuscript again and corrected some typing mistakes, which we apologize for.
Please, the introduction should be reformulated in a better way, in order to provide both, a correct and essential background of the topic and to permit the readers to clearly understand the contents.
Authors’ answer: Thank you for your suggestion. We have added information to try to improve the rationale of the introduction.
The section 2 should be the Materials and Methods section.
Authors’ answer: We apologize for this mistake. We have changed the title of the section.
At line 103, the authors state that a pilot study was performed. Please, include the reference to provide the results of the pilot study.
Authors’ answer: Thank you for your constructive criticism. We have better explained the development of the questionnaire. This information is in lines 117-124.
Results
In the section 3.1, authors reported the Mann Whitney test in an incorrect manner. Please, report the results in the correct way, following an existing style.
At the same way, the chi-square test was incorrectly reported. Please, write the chi-square results in the correct manner.
Authors’ answer: Thank you for your constructive criticism. We have rewritten the reports on the Mann Whiney and Chi square tests.
The description of Figure 1 is missing. Please, the authors should explain better what the figure reports and the meaning of the asterisk.
Authors’ answer: Thank you for your suggestion. The description of Figure 1 is at the bottom of the image. We have deleted the asterisk as we have now reported the results using another pattern.
In the following part of the results section, the authors report the chi-square results in an incoherent manner. Please, uniform the style used to report statistical results.
Authors’ answer: Thank you for your constructive criticism. We have rewritten the reports on the Mann Whiney and chi square test.
In section 4.3, the authors used interchangeably the concept of exercise and physical activity, while they are different.
Authors’ answer: Thank you for your suggestion. We have reviewed the terms physical activity and exercise and we agree that we have not used them correctly. This content is now focused on physical activity not on exercise. We have also added the Physical activity recommended by the World Health Organization in lines 94-99.
In addition, please less speculation should be done on results unsupported by data.
Authors’ answer: We agree with the reviewer and have deleted the statement unsupported by data.
The remaining part of the manuscript is poor written and includes typing errors. Please, reformulate in a correct manner.
Authors’ answer: We have reviewed the content and the English and sent it to a professional translator to be proofread.
Finally, in the affiliation section, the instruction provided from the editor are not respected.
Authors’ answer: We are not clear about this criticism. We added an asterisk (*) to the corresponding author’s name. We want to clarify that “López de Subijana” is one surname and that “L.” is Jose L. Chamorro’s middle name.

Round 2
Reviewer 2 Report
The authors made the revision in an incorrect manner. There are even errors that were not present in the previous version.
For example, the first sentence of the introduction was copied in the abstract with an attached reference. Furthermore, the revised version lacks the author's contribution, funding and conflict of interest parts.
In the introduction section, authors are not coherent about the purpose of the study.
The authors state that a professional revision of the language has been done. However, the quality of English is still not satisfactory, affecting the sense of the article.
The authors enter information in the wrong sections (e.g., statistical results in materials and methods rather than in the results section).
The authors did not use a consistent way to report statistical results, and they did not describe the results correctly. For example, the Mann U Whitney is used to assess if differences between two groups exists, taking into account the samples mean. The authors do not consider the rationale of this test, describing the results in a misleading way (e.g., Line 169-171).
Overall, the article has errors regarding form and content. In addition, the authors incorrectly report the results of the study.
Accordingly, the article is rejected.
Author Response
REVIEWER 2
"The authors made the revision in an incorrect manner. There are even errors that were not present in the previous version.
Authors’ answer:
Dear editors and reviewer:
We really appreciate the effort you made in reviewing our manuscript for the second time. We want to apologize as it seems that the changes we made did not reach the quality required.
We have addressed each comment and we have highlighted the changes in red. The three rounds of the English version review of our manuscript have been proofread by a professional English Translator, and we attached a certificated from her. We hope this version is suitable for publication.
Thank you for everything, best regards,
The authors
------------------------------------------------------------------------------
For example, the first sentence of the introduction was copied in the abstract with an attached reference.
Authors’ answer: We have modified the text and erased the reference.
Furthermore, the revised version lacks the author's contribution, funding and conflict of interest parts.
Authors’ answer: We have added again this part as in the original version. It is now written in lines 288-295.
In the introduction section, authors are not coherent about the purpose of the study.
Authors’ answer:. To make the the purpose of the study more clear we have added and retirement to the title and the relevant sections 114-115. The structure of the introduction is focused, firstly, to describe the relevant theoretical models used in career development and transitions research (Stambulova &Wylleman, 2019, doi:10.1016/j.psychsport.2018.11.013). Secondly, facilitators and barriers that athletes must face in their career developments. Finally, we present why there could be differences in career development and retirement depending on the type of sport. In our opinion, all the information used in the introduction section is relevant for understanding the purpose of the study.
The authors state that a professional revision of the language has been done. However, the quality of English is still not satisfactory, affecting the sense of the article.
Authors’ answer: We are non-native English speakers, so the English reviewing process was done by a professional translator who is a native English speaker with more than 40 years’ experience. We attached a certificate regarding our final version of this manuscript.
The authors enter information in the wrong sections (e.g., statistical results in materials and methods rather than in the results section).
Authors’ answer: Thank you for the constructive criticism. We have removed that sentence as suggested by the reviewer.
The authors did not use a consistent way to report statistical results, and they did not describe the results correctly. For example, the Mann U Whitney is used to assess if differences between two groups exists, taking into account the samples mean. The authors do not consider the rationale of this test, describing the results in a misleading way (e.g., Line 169-171).
Authors’ answer: Thank you for the comment. We agree that the Mann Whitney U test was not reported correctly. We have reviewed again the content and we have followed the same pattern for each one. Now the content is based on the mean of each variable. Regarding the Pearson Chi square test, we did change the way of presenting the results. First, we rewrote the general comparison and second, we analyzed the adjusted standardized residuals in the cases where dependent variables have more than two answers. The analysis of these residuals is based on the number of columns and rows. Our two reported results are 4x2 = 8 number of cells. In order to interpretate the α .05, it should by divided by the number of cells. Then .05/8=.00625 which leads us to a z of ± 3. The adjusted residuals reported in our manuscript are above 3 so there were significant differences at that level of the variable between both groups of the independent variable.
Please see:
- Agresti, A. (2002). Categorical Data Analysis (2nd Ed.). New York: Wiley. (p. 38; p. 81)
- Field, A. (2013). Discovering statistics using SPSS (4th ed.). London, UK: Sage.(p743-747).
- MacDonald, P. L., & Gardner, R. C. (2000). Type 1 error rate comparisons of post hoc procedures for I J chi-square tables. Educational and Psychological Measurement, 60, 735-754. doi.org/10.1177/00131640021970871
Therefore, we have rewritten the statistical analysis in lines 158-161.
Overall, the article has errors regarding form and content. In addition, the authors incorrectly report the results of the study.
Authors’ answer: Thank you for your comment. We have modified the way of reporting the Mann-Whitney U test and we have provided supporting references for the way of reporting the Pearson Chi-Square test.